# China Requires a Sustainable Transition of Vegetable Supply from Area-Dependent to Yield-Dependent and Decreased Vegetable Loss and Waste

**DOI:** 10.3390/ijerph20021223

**Published:** 2023-01-10

**Authors:** Ying Tang, Jinlong Dong, Nazim Gruda, Haibo Jiang

**Affiliations:** 1Department of Agronomy and Horticulture, Jiangsu Vocational College of Agriculture and Forestry, Jurong 212400, China; 2State Key Laboratory of Soil and Sustainable Agriculture, Institute of Soil Science, Chinese Academy of Sciences, Nanjing 210008, China; 3Institute of Crop Science and Resource Conservation, Division of Horticultural Sciences, University of Bonn, 53121 Bonn, Germany; 4Jiangsu Station for Protection of Arable Land Quality and Agricultural Environment, Nanjing 210029, China

**Keywords:** dietary pattern, food loss and waste, model prediction, vegetable supply pattern, vegetable yield

## Abstract

China, the largest country in vegetable supply, faces a transition to sustainable vegetable production to counteract resource waste and environmental pollution. However, there are knowledge gaps on the main constraints and how to achieve sustainable vegetable supply. Herein, we integrated the vegetable production and supply data in China, compared its current status with five horticulture-developed countries US, the Netherlands, Greece, Japan and South Korea, using data from the Food and Agriculture Organization (FAO) and National Bureau of Statistics of China, and predicted the vegetable supply in 2030 and 2050 by a model prediction. The vegetable supply in China increased from 592 g capita^−1^ d^−1^ in 1995 to 1262 g capita^−1^ d^−1^ in 2018 and will keep constant in 2030 and 2050. Compared to the five countries, the greater vegetable supply is primarily achieved by higher harvested areas rather than higher yield. However, it is predicted that the harvested areas will decrease by 13.6% and 24.7% in 2030 and 2050. Instead, steady increases in vegetable yield by 11.8% and 28.3% are predicted for this period. The high vegetable supply and greater vegetable preference indicated by the high vegetable-to-meat production ratio cannot guarantee recommended vegetable intake, potentially due to the high rate of vegetable loss and waste. Under the scenarios of decreased vegetable loss and waste, the harvested area will decrease by 37.3–67.2% in 2030 and 2050. This study points out that the sustainable transition of Chinese vegetable supply can be realized by enhancing yield and limiting vegetable loss and waste instead of expanding the harvested area.

## 1. Introduction

In the recent four decades, vegetable supply in China has substantially improved, as indicated by sharp increases in both the total quantity of vegetable supply and the harvested area [1]. China, therefore, has become the largest country for vegetable production and has a remarkably high vegetable supply per capita worldwide [2,3]. With the fast economic development, governments and vegetable producers gradually increased their investment in vegetable production to enhance farmers’ income as a measure of anti-poverty for smallholder farmers in particular and increase vegetable availability for a better living standard [4]. The expanded vegetable industry and, thus, the high vegetable supply also improve vegetable affordability and facilitates the vegetable intake and a healthy diet in China compared to other countries [5].

However, the significant investment into the vegetable industry and thus increased harvested area and supply quantity have resulted in severe resource waste and environmental pollution [3,6,7]. The remarkable increase in vegetable supply quantity is demonstrated to be far above vegetable requirement by people in some specific regions at some time, leading to a significant decrease in vegetable prices [6,8], and resulting in serious vegetable loss and waste, vegetable abandoned without harvest in the field as an example. In addition, some smallholder farmers with little experience in vegetable cultivation and marketing would prefer to take advantage of the investment by governments to commence vegetable cultivation. However, they cannot manage the facility or soils appropriately in the long run, leaving the land or facility discarded [9]. Concerning environmental pollution, vegetable cultivation has been widely demonstrated to increase greenhouse gas emission [4], nonpoint source pollution, and underground water pollution, resulting in low nutrient use efficiency [10] and the degradation of soil quality [3,9]. Therefore, the dramatic expansion of vegetable-harvested areas keeps threatening the environment of agricultural ecosystems [3].

Hence, China is urgently required to have a transition to guarantee resource-saving and environment-friendly systems for sustainable vegetable supply [3,10]. However, the knowledge gaps on the main constraints of achieving sustainable vegetable supply still exist, and it is unclear how to take some specific measures to advance the sustainable transition of vegetable supply in China. This study aims to fully understand the current status of vegetable supply, compare it with horticulture-advanced countries in vegetable supply, and predict the future of the vegetable supply in China. We believe that this knowledge will undoubtedly improve our understanding of managing vegetable cultivation in China sustainably. Furthermore, it will facilitate the government for policymaking and vegetable growers, particularly smallholder farmers, for future successful management of vegetable farms.

## 2. Materials and Methods

### 2.1. Parameter Definition

The definition of a vegetable is followed by the Food and Agriculture Organization (FAO), excluding tubers, potatoes, non-green legumes and pulses. In addition, vegetables grown principally for animal feed or seeds are excluded. The definition of vegetable supply is the subtraction of vegetable production and imports to exports and changes in stocks [11]. The vegetable intake in this study is the intake of fresh vegetables, neglecting the processed vegetables that were taken up in many low amounts [12], e.g., tomato juice or dehydrated vegetables. We define the vegetable loss and waste ratio (%, Y_vegetable loss and waste_) as the ratio of the subtraction of vegetable intake (X_intake_) from the vegetable supply (X_supply_) to vegetable supply (Equation (1)). The definition of vegetable loss and waste is adapted to the FAO definition of food (vegetable) loss and waste: vegetable loss refers to the decreases in the quantity or quantity of vegetables by food suppliers that occur at the stages of production, postharvest, storage, and processing, excluding retailers, food service providers and consumers; vegetable waste refers to the quantity or quantity of vegetable discarded at the stage of distribution, retailing and consumption by retailers, food service providers and consumers [13,14]. Therefore, the vegetable loss and waste potentially include harvest, postharvest and storage waste, transportation and processing waste from field to market, vegetable preparation waste in the kitchen, and not-eaten vegetables on the table. Since the data on the intake of vegetables excluding fresh vegetables, e.g., processed or dehydrated vegetables, was not available and thus not considered in the current equation despite being small portions, the ratio of vegetable loss and waste might be overestimated to an extent. We define the vegetable –meat ratio as the ratio of the supply of vegetables to those of meat plus fish to indicate the vegetable preference of customers.
(1)$$Y_{\mathrm{vegetable}\;\mathrm{loss}\;\mathrm{and}\;\mathrm{waste}}=\frac{X_{\mathrm{supply}}\;-\;X_{\mathrm{intake}}}{X_{\mathrm{supply}}}\;\times \;100\%$$

### 2.2. Data Source

The data on China from 1995 to 2018 were collected from the National Bureau of Statistics of China [1] and the FAO database. The data comprised the total population, gross domestic product (GDP), production quantity, vegetable intake per capita, vegetable yield, area harvested per country, and export quantity of vegetables. The definitions of this parameter were adopted from FAO directly [11]. In addition, we selected five typical countries of vegetable production to compare with China to outline the status and to predict the potential roadmap for developing the vegetable supply system in China. United States of America (US) and the Netherlands are characterized by high vegetable yield, high meat intake, and high GDP associated with advanced horticultural technology [15,16]. Besides these, the US imports numerous vegetables whilst the Netherlands is famous for vegetable exportation. Japan and South Korea are geographically close to China and have had similar dietary patterns historically [16]. South Korea and Greece are two specific countries with high vegetable supply and intake, similar to China [17,18]. Besides the data mentioned above, the supply of vegetables, meat and fish from 1979 to 2018 in all six countries and the vegetable import from China were collected from the FAO database [19].

### 2.3. Model Prediction

This study assumed that the vegetable supply (production) could be improved and then kept constant when the economic development of a country, as indicated by GDP, increased. The response curve that the vegetable supply of China, Greece, and South Korea (Y_Supply_) increased with GDP (X_GDP_) and then kept constant confirmed our assumption. This assumption has also been confirmed by some studies [15,16]. Then, the linear fitting, sigmoidal fitting, and polynomial fitting were conducted and found that the sigmoidal curve fitted better by using the Function of Doseresp (Equation (2)) in OriginPro 8 SR0 (v8.0725). The sigmoidal curve fittings were conducted for South Korea and Greece due to their high and similar vegetable supply compared to China.

In addition, we predict the vegetable supply in China in the near future, i.e., 2030 and 2050, by a sigmoidal curve fitting with the Function of Doseresp using the data of vegetable supply response to time scale from the National Bureau of Statistics of China, so made the prediction of vegetable intake. We also predicted the vegetable yield in 2030 and 2050 based on the linear correlation (R^2^ = 0.906, *p* < 0.001) between vegetable yield and time. The vegetable supply to vegetable production ratio was taken as a coefficient of 0.92 in 2018 based on the FAO database, achieving the calculation of vegetable production. Hence, the area harvested in 2030 and 2050 was calculated by dividing vegetable production by vegetable yield.
(2)$$Y_{\mathrm{Supply}}=\;A1+\frac{A2\;-\;A1}{{1+10}^{(\mathrm{LOGX}0\;-\;X_{\mathrm{GDP}})\;p}}$$ 
A1, A2, and LOGX0 are constants.

We used the same principle to predict the area harvested under various vegetable loss and waste scenarios. Briefly, the vegetable loss and waste ratios of 70%, 60%, and 50% were utilized to multiply the predicted vegetable supply when a plateau was reached, and then calculated vegetable production by using the coefficient of 0.92. The area harvested in 2030 and 2050 was calculated by dividing vegetable production by vegetable yield.

## 3. Results

The vegetable supply in China gradually increased from 592 in 1995 to 1262 g per capita^−1^ d^−1^ in 2018, increasing by 113% (Figure 1). Whilst it is unexpected to observe that the vegetable loss and waste ratio showed a similar trend to vegetable supply, the ratio increased from 47% in 1995 to 79% in 2018. Vegetable intake increased from 313 in 1995 to 331 g capita^−1^ d^−1^ in 2002, increasing by 5.6%, above the recommended intake level of 300 g capita^−1^ d^−1^ of the EAT-Lancet Commission [20] and comparable to 300–500 g capita^−1^ d^−1^ of Chinese Dietary Guidelines [21]. However, vegetable intake decreased to 263 g capita^−1^ d^−1^ in 2018, by 16% compared to the Year 1995. The slopes of the increase in vegetable supply, vegetable loss and waste ratio, and area harvested per capita and per country in 2002 was greater than other parameters. The sharp increase in GDP per capita after 2002 was associated with decreased vegetable intake. The import, export and yield were linearly correlated with the year, whilst the import and export accounted for only 0.2% and 1.3% of total vegetable production in 2018 in China. 

Compared with the other five horticulture-advanced countries, China had the lowest vegetable yield and lowest GDP as an indicator of low customer income and low horticultural technology. However, the vegetable supply, area harvested per capita and per country, and vegetable meat ratio in China was the greatest in recent two decades (Figure 2). The vegetable supply and area harvested per capita and per country in China kept increasing in the past four decades whilst they increased to a lesser extent from the Year 2015. By contrast, these parameters were either kept constant or decreased to an extent in the other five countries. Similarly, the vegetable meat ratios in South Korea and Greece decreased whilst those in the US, Japan, and the Netherlands kept constant in recent three decades. The increases in yield by year were always observed across all six countries, whilst the yield in Japan and South Korea increased more slowly or even decreased in the recent ten years. Compared with the US, the Netherlands, and Japan, the vegetable supply, area harvested per capita, and vegetable-to-meat ratio in China were closer to South Korea and Greece from 1985 to 2005. This, in general, indicates the similarities in vegetable preference among China and South Korea and Greece.

Regardless of the National Bureau of Statistics of China or FAO data, the curve fitting showed that vegetable supply across China, South Korea, and Greece increased by increased GDP per capita and kept constant roughly from 2500 US$ capita^−1^ (Figure 3 and Table 1). Vegetable supply in China peaked at 1219.0 g capita^−1^ d^−1^ based on data from the National Bureau of Statistics of China and peaked at 929.1 g capita^−1^ d^−1^ based on FAO, whilst those of South Korea peaked at 681.1 g capita^−1^ d^−1^, and Greece peaked at 565.2 g capita^−1^ d^−1^. 

Our model prediction showed that the vegetable supply and intake in China kept constant at 1219 and 265 g capita^−1^ d^−1^, respectively, in either 2030 or 2050 since 2018, whilst vegetable yield and area harvested were 38.5 t ha^−1^ and 125.7 m^2^ capita^−1^ in 2030, and 44.1 t ha^−1^ and 109.5 m^2^ capita^−1^ in 2050, respectively, from 34.4 t ha^−1^ and 145 m^2^ capita^−1^ in 2018 (Figure 4). The vegetable yield will increase by 11.8% and 28.3% in 2030 and 2050, respectively. The vegetables harvested area decreased by 13.6% and 24.7% in 2030 and 2050, respectively, compared to those in 2018.

When the vegetable loss and waste ratio decreased from 79% in 2018 to 70%, 60% and 50%, the vegetable supply would have decreased to 885, 664, and 531 g capita^−1^ d^−1^, respectively, achieving 265 g capita^−1^ d^−1^ vegetable intake. In this case, the area harvested per capita in 2030 decreased to 91, 68 and 55 m^2^ capita^−1^, and those in 2050 decreased to 79, 60, and 48 m^2^ capita^−1^, respectively. The area harvested in 2030 decreased by 37.3%, 53.0%, and 62.4%, whilst those in 2050 decreased by 45.3%, 59.0%, and 67.2%, respectively.

## 4. Discussion

### 4.1. From Area-Dependent to Yield-Dependent Vegetable Supply

Our study demonstrated that China is under the transition from vegetable harvested-area-dependent to yield-dependent (Figure 5). Compared to the other five horticulture-developed countries, the greater harvested area per capita or per country but the lower vegetable yield in China indicate that high vegetable supply was achieved by expanding harvested area more than the yield before 2018 (Figure 2b, c). This conclusion is supported by the close linear correlation between vegetable supply and harvested area per capita (R^2^ = 0.964, data not shown). However, this condition was gradually changed since 2003, as indicated by the slight increase in harvested area per capita and an increase in yield (Figure 1b). Also, based on our model prediction, the contribution of vegetable yield to vegetable supply would increase whilst that of area harvested per capita decreased by 13.6% and 24.7% in 2030 and 2050, respectively (Figure 4a). The vegetable farms in China were, as a general, kept concentrated in some specific regions, such as in North China and Bohai Rim, which accounts for ca. 30% of the Chinese vegetable harvested area. This concentrated vegetable cultivation facilitates the advanced technology being increasingly invested and utilized in vegetable production, thus enhancing vegetable yield [9,22]. This strategy has been proven successful in other horticulture-developed countries in Italy and the Almeria of Spain [23]. In this case, the harvested area would be gradually shrunk, leading to more technology-concentrated and easily managed horticultural farms that can be expected to be resource-saving and environment-friendly [3]. 

Therefore, the policymakers in China should avoid more investment or subsidies to enlarge cultivation areas for vegetable production in the future as they did previously as an effective measure to liberate poverty [24]. The fast expansion of harvested areas by farmers with little cultivation experience has been threatening the local environment, indicated by a higher greenhouse gas emission of 49.2% [25], a higher nitrate accumulation in soils of 258% [9] and thus polluting either underground or surface water [10]. Low nutrient use efficiency of 18.6% for nitrogen under the condition of plastic-shed vegetable production is an example [26]. In addition, the high affordability of vegetables has been cracking the confidence of smallholder farmers in some specific regions due to the low vegetable price from the intensive competition [6]. Instead, more investment in controlled environmental agriculture and intensive and sustainable technologies, such as soilless culture systems [27,28], could support a high-yield and high-quality transformation of vegetable production.

### 4.2. Decreased Vegetable Loss and Waste Are Required

On the other hand, the increased vegetable supply did not facilitate vegetable intake in China. Our results showed that vegetable intake increased with increased vegetable supply before 2002 whilst vegetable intake decreased despite increased vegetable supply after 2002 (Figure 1a). The vegetable loss and waste ratio superficially demonstrated that instead of increasing vegetable intake, the increased vegetable supply contributes to the high vegetable loss and waste, as indicated by the high ratio (79%) (Figure 1 and Figure 5). The vegetable loss and waste ratio are increasing with increasing vegetable supply across various countries worldwide [5]. Based on their data, we calculated that the vegetable loss and waste ratio in China was 18% in 2018, confirming our study. This result is further confirmed by another study that demonstrated the loss and waste ratio of vegetables and fruits in total was ca. 50% in China [13]. Since vegetables are easier to be wasted than fruits, leafy vegetables in particular, we can expect a similar ratio of loss and waste of vegetables with our result. The vegetable producers and policymakers support the expansion of vegetable-harvested areas, and thus, increased vegetable supply appears to be less effective in motivating vegetable intake compared to increasing vegetable loss and waste. This is undoubtedly a tragedy despite higher vegetable availability making vegetables more affordable in China.

We speculate that the urban lifestyle increases vegetable loss and waste and changes vegetable supply systems. Firstly, older adults prefer or get used to a vegetable diet, whilst the young generations prefer having more meat, leading to more vegetable loss and waste in the kitchen, particularly when vegetable foods are prepared by elders [29,30]. In addition, the culture of social interaction in restaurants, large portion sizes of served food, having vegetable dishes at the end of a meal in China, and an increasing number of small family sizes strengthen vegetable loss and waste to some extent [13]. Secondly, the excessive labor force in the countryside during urbanization, elderly farmers, in particular, have experience in crop cultivation and thus would prefer being involved in vegetable production when they cannot find appropriate jobs locally. This results in excessive vegetable production far over the requirement, leading to vegetable waste due to low prices and thus unharvested in some sites [31]. Lastly, the systems of vegetable supply prolong the distance and time consumption between vegetable production and intake, leading to more vegetable loss and waste during transportation or marketing of vegetables, leafy vegetables in particular (Figure 5) [30,32]. Our model prediction showed that the area harvested could decrease by 37–62% in 2030 when the current waste rate decreases from 79% to 70–50% and the decrease in the area harvested could be greater in 2050 (Figure 4). It indicates that a loss-and-waste controlled strategy will result in more arable land savings and, thus, sustainable vegetable production. 

We found that the vegetable –meat ratio increased and then remained constant during the recent three decades in China (Figure 2d), showing that people’s preference to eat vegetables in their daily diet did not increase. While people’s preference to eat vegetables kept decreasing in the horticulture-advanced countries, China might face decreased vegetable preference in the future, further decreasing vegetable intake. The below-recommended level of vegetable intake has been a severe issue for human health across the world [15,20]. The fact that even the super-high vegetable supply and high vegetable preference in China cannot guarantee sufficient vegetable intake further warns people to attach more importance to the dietary pattern, supply chain of vegetables and loss and waste management (Figure 5). It has been demonstrated to be less successful by only emphasizing the health benefit of vegetables [33,34]. Therefore, it is time to reshape the urban lifestyle and advocate healthy dietary patterns. For example, children are urgently required to interact with mother nature face to face to bond with vegetable plants, which potentially buries vegetables in mind as a living body more than as foods, thus increasing the attractiveness and intake of vegetables [35]. In addition, the current vegetable supply system should improve vegetables’ tastes by increasing vegetable diversity with new species or cultivars, cooking styles or postharvest processing to enhance vegetable preference [36,37]. Considering the changes in our climate and the increasing responsibility of consumers in this respect, we strongly suggest that intervention toward increasing vegetable preference and the reduction of meat consumption are required, which is healthy and environment-friendly [38,39].

### 4.3. Uncertainty

The current model is under the scenario that the vegetable supply and production are mainly market-orientated without much social instability. However, in food production in China, many people are more or less policy-motivated. Therefore, food availability and, thus, security are considered by policymakers in the long run. Therefore, vegetable production and supply might be strengthened to ensure sufficient food supply under a state of emergency in the future, such as COVID-19 [40], which decreases the prediction accuracy. In addition, the model prediction did not consider much about the agronomic and environmental (such as climate changes) factors that affect yield and vegetable cultivation despite the time trend of vegetable yield being perfectly linear. Therefore, the readers should be careful when considering the application of the current model prediction.

The data from the National Bureau of Statistics of China and FAO are different to a small extent, indicating that the accuracy of the data can be a challenge. The data collection of the harvested area is the most challenging as farmers or producers might overestimate their harvested area due to the wants of the subsidy from the government. The over-estimated harvested area can result in greater total vegetable production and, thus higher ratio of vegetable loss and waste to an extent.

## 5. Conclusions

This study demonstrated that the vegetable supply in China gradually increased from 592 g capita^−1^ d^−1^ in 1995 to 1262 g capita^−1^ d^−1^ in 2018 and is predicted to keep constant in the future. The greater vegetable supply was achieved by higher harvested area than yield when compared to technology-advanced countries currently. The harvested areas are predicted to decrease in 2030 and 2050 due to the increases in vegetable yield. However, the greater vegetable supply cannot guarantee sufficient vegetable intake, implying a high rate of vegetable loss and waste. The harvested area can be decreased by 37–67%, as predicted in 2030 and 2050 when vegetable loss and waste decrease. The predicted decrease in the harvested area would contribute to the transition of vegetable supply systems to be more resource-saving and environment-friendly. This study warns policymakers to avoid any reckless actions to enlarge the cultivation area for vegetable production in China. As vegetable intake is currently deficient, practical measures to change the urban dietary pattern or to improve the vegetable supply chain should be taken toward more sustainable vegetable production.

## Figures and Tables

**Figure 1 ijerph-20-01223-f001:**
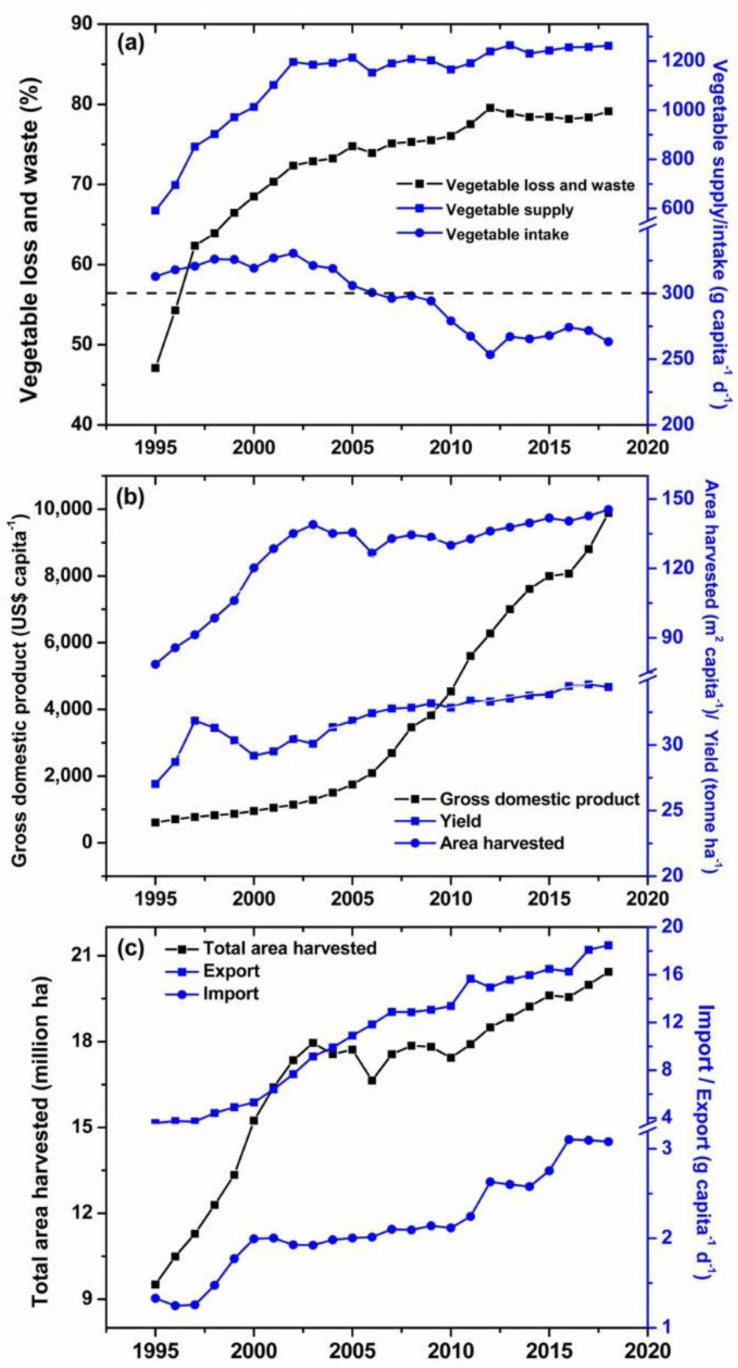
The supply, intake, loss and waste (**a**), area harvested per capita, yield, area harvested per country (**b**), import and export of vegetables, and gross domestic product (**c**) in China from 1995 to 2018. Data are collected from the National Statistics of China except for the data on vegetable import that originated from the FAO database. The dot-line in Figure 1a represents the recommended value of vegetable intake by the EAT-Lancet Commission [20].

**Figure 2 ijerph-20-01223-f002:**
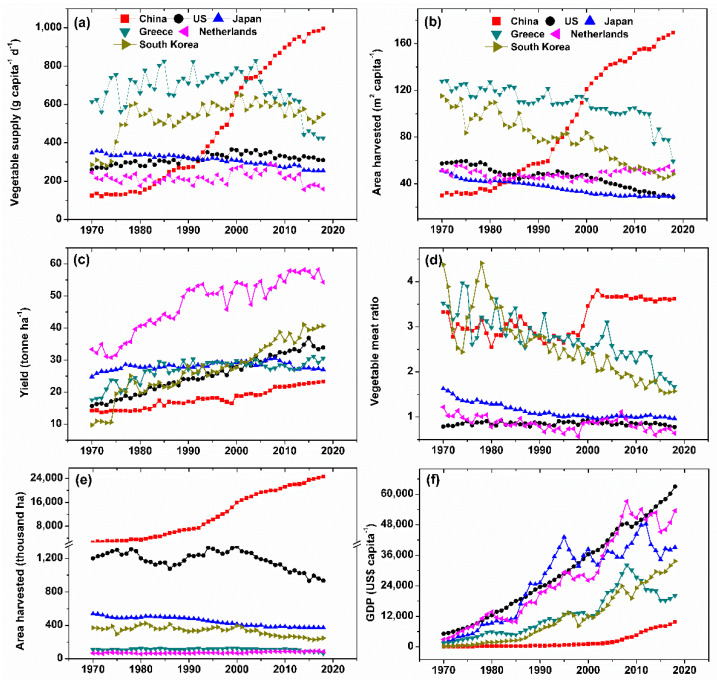
The vegetable supply (**a**), area harvested per capita (**b**), yield (**c**), vegetable to meat ratio (**d**), total area harvested per country (**e**), gross domestic product (GDP) (**f**) of China, United States, Japan, Greece, Netherlands, and South Korea from 1970 to 2018. Data originated from the FAO database. The vegetable–meat ratio is the ratio of the quantity of vegetable supply to that of meat plus fish.

**Figure 3 ijerph-20-01223-f003:**
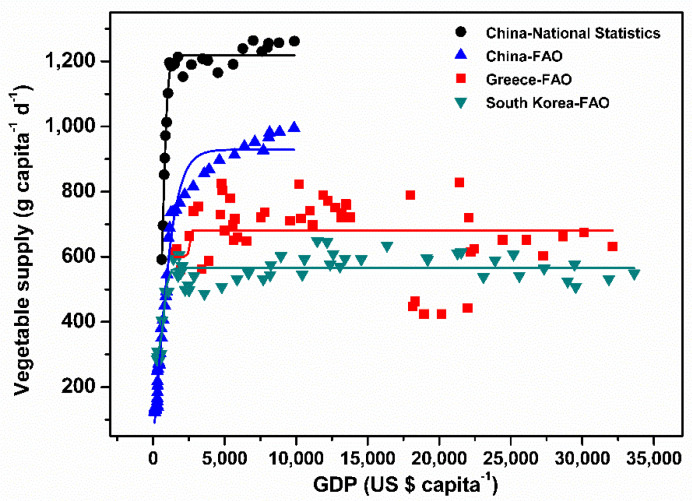
The sigmoidal curve fittings between gross domestic product (GDP) and vegetable supply are based on the data from the National Statistics of China from 1995 to 2018. The data from South Korea and Greece originated from the Food and Agriculture Organization (FAO) database from 1970 to 2018. The *p*-value of the curve fitting of China and South Korea is <0.001. The trend of Greece, as indicated by the fitting curve, is shown despite being not significant.

**Figure 4 ijerph-20-01223-f004:**
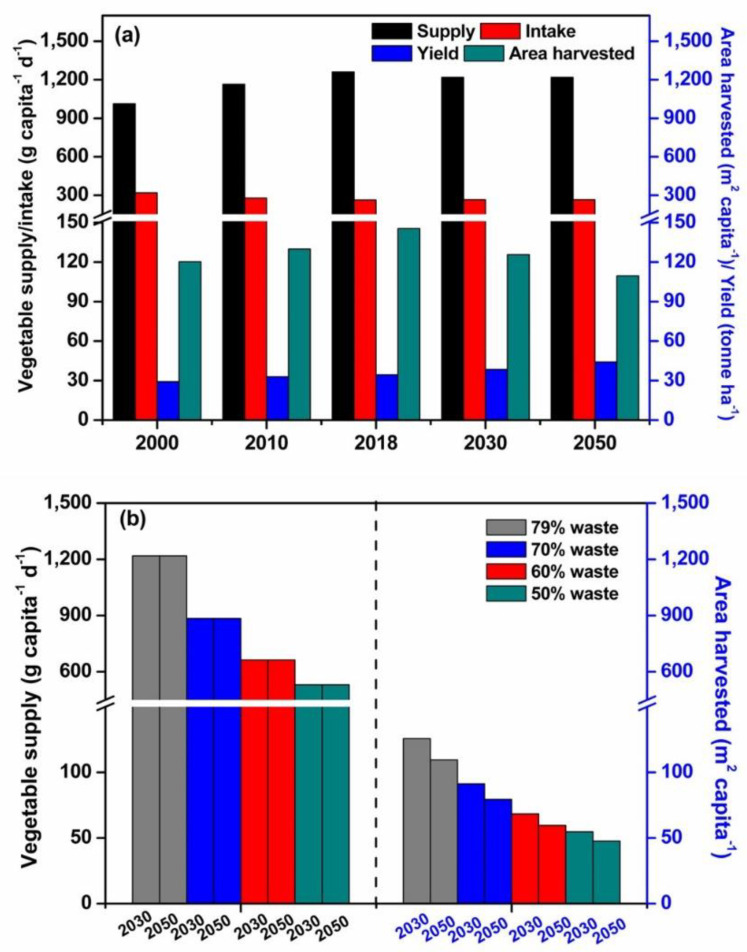
The supply, intake, area harvested per capita, and yield of vegetables in China in 2000, 2010, 2018, and the predicted 2030 and 2050 (**a**), the vegetable supply and area harvested per capita considering the various scenarios of vegetable loss and waste of 70%, 60%, and 50%, as predicted in 2030 and 2050 (**b**). Data are collected from the National Statistics of China.

**Figure 5 ijerph-20-01223-f005:**
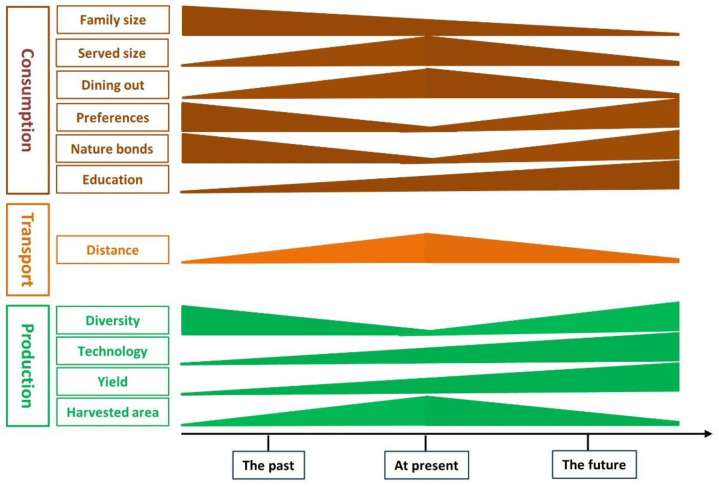
The production, transportation, and consumption of vegetables at three-time scales (the past, at present, and the future) in China. Vegetable production is gradually shifted from area dependent to yield-dependent by using advanced technology. The loss and waste of vegetables are increased due to the excessive cultivation area, long transportation, small family size, and decreased people’s preference for vegetables. The increase in vegetable intake, together with less loss and waste in the future, can be achieved by more education on the vegetable diet from either a health or environmental perspective, closer bonds with vegetable plants and nature, increased vegetable diversity, shortening the distance of vegetables from the fields to tables, modified dietary pattern (smaller served size and less waste on tables from dining out), and using modern technology.

**Table 1 ijerph-20-01223-t001:** The values of parameters of the sigmoidal curve fitting between gross domestic product (GDP) and vegetable supply are shown in Figure 3.

Parameters	Meaning	China National Statistics	China FAO	South Korea
$A_{1}$	Bottom asymptote	278 ± 242	−613 ± 397	271 ± 42
$A_{2}$	Top asymptote	1218.9 ± 8.5	929 ± 12	565.2 ± 6.4
$LOGx_{0}$	Centre	722 ± 87	234 ± 338	732 ± 81
*p*	Hill slope	0.00276 ± 0.00058	0.00065 ± 0.00010	0.00294 ± 0.00141
R^2^	Coefficient	0.968	0.984	0.808

Values are predicted values ± se.

## Data Availability

All the data can be downloaded from corresponding website or obtained upon request from the authors.

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
