# Peer review of "China Requires a Sustainable Transition of Vegetable Supply from Area-Dependent to Yield-Dependent and Decreased Vegetable Loss and Waste"

_ijerph, 2023, doi:10.3390/ijerph20021223_

Round 1
Reviewer 1 Report
Thank you for your work. This paper is generally well-written and the narrative flow is good. The manuscript seeks to present and then model how vegetable consumption/production patterns in China change. However, I have several hesitations for recommending this for publication, which are mostly centred around a need for a greater critical discussion of the limitations of the modelling approach, a more critical comparison to the literature, and a better/more quantitative description of the results. My comments are listed below.
Names: I note that the primary author has a ‘1’ attached to their name. This is perhaps a typographical error.
Abstract
Line 20: This notation is difficult to read, please change to using a slash instead or superscripts: e.g. 592/capita/day is much more naturally and easily read than 599 capita -1 day -1.
21: what was basis of selection?
23: predicted how? What method?
79 and 94 defines à ‘define’
98 ‘or’ à is this meant to be ‘and’?
Section 2.2 defines the criteria for comparison to specific set of 5 countries. They seem to be fair criteria but I would strongly suggest that this be defined better, ie through references and data rather than these broad and subjective-sounding statements.
Results section is difficult to read because of the notation. I see it is possible for you to add subscripts/superscripts as with the previous paragraph and your equations. Please improve this section’s readability
141-142 what does more significant mean? What type of statistical tests were used and how was significance defined?
Figure 2 please place the legend in larger size beside the six graphs for easier readability rather than where it is currently placed only in one corner of one graph.
143-144 ‘import’ and ‘export’ need to be more clearly defined, and I don’t see these percentages in the graph as described.
159-163 these lines need to be more quantitatively described. At present they are very quantitatively described, eg what is steady? What is ‘slowly’? describe these figures in text rather than relying on the reader to figure them out in the graphs.
164-166 ‘Some similarities’ with regards to what? Too broad. Revise, or make more detailed, add references if needed.
175 again, what is ‘steady’? Describe this including the figure.
176 please see my comments about notation.
212-213 “slight” and “steady” descriptors
Figure 5 Where is this figure from? I would not include it in the article. Perhaps it is more suitable for a presentation but not in a scientific manuscript.
218 where is this production data from? This is interesting and should be shown in the paper (eg production/supply graph) if it is referred to here.
234-236 This is a rather unfounded statement, some references needed here.
249 there is insufficient evidence in the paper to make this assumption about preference, which is not thoroughly addressed in the paper.
Section 4.1 Beyond uncertainty I would strongly suggest the limitations of the paper be discussed here. Making projections based solely on a time trend (sigmoidal or not), does not account for the agronomic, environmental (climate change, land use) factors that affect yield, eg the G x E x M “formula” (genetics, environment, management) for yield and cultivation. Significant energy should be given to think critically about these limitations as well as their relevance compared to other similar studies if necessary.
Author Response
Dear Editor,
We very much appreciate the comments from reviewers to our manuscript. All comments and suggestions have been carefully considered revised in red in the text. Our responses to individual comments are outlined in the attachment. Hope the revised version becomes acceptable for publication.
Yours sincerely,
The authors

Reviewer 2 Report
Please see attached.

Author Response

(The authors gave the same response as above.)

Round 2
Reviewer 1 Report
Thank you for improving your work. It reads much more clearly and better and the statistical analysis has been improved as well. However, I still have reservations about the suitability of Figure 5 for a scientific journal and would still recommend its removal.
Author Response
We understand your worry that having Figure 5 for a scientific journal appears to be less scientific. However, we see the tendency of various well-known journals having a figure of the framework of their papers or graphic abstract to attract the readers' interests and to make readers easy to understand their contents before getting into the long wording. This is particularly useful when readers are not in this field or scientists. Therefore, we spent 500 US$ to make this figure achieving this target. We sincerely hope you could understand what we think, and allow us to have this figure.